

# In situ cloud surface measurements dataset from four cloud spectrometers during the Pallas Cloud Experiment (PaCE) 2022.

Konstantinos Matthaios Doulgeris[1], Ville Kaikkonen[2], Harri Juttula[2], Eero Molkoselkä[3], Anssi Mäkynen[3] and David Brus[1]

[1]Atmospheric Composition Research, Finnish Meteorological Institute, Helsinki, 00100, Finland
[2] Unit of Measurement Technology, University of Oulu, Technology Park, Kajaani,87400, Finland
[3]Optoelectronics and Measurement Techniques research unit, University of Oulu, Erkki-Koiso Kanttilan katu 3, Oulu, 59700, Finland

*Correspondence to*: Konstantinos M. Doulgeris (Konstantinos.doulgeris@fmi.fi)

**Abstract.** This data paper presents an overview of the cloud spectrometers deployed during the Pallas Cloud Experiment (PaCE) in autumn 2022, a coordinated measurement campaign in the Finnish subarctic that took place between September 12th and December 15th, 2022. Four cloud spectrometers, the Cloud Aerosol Spectrometer (CAS), the Forward Scattering Spectrometer Probe (FSSP-100), the Cloud Droplet Analyzer (CDA), and the ICEMET were operated as ground-based setups, providing high-resolution, in-cloud measurements of droplet size distributions and key microphysical properties, such as number concentration ($N_c$), liquid water content (LWC), median volume diameter (MVD), and effective diameter (ED). The dataset is complemented by meteorological observations of temperature, humidity, wind speed, and visibility at a 1 min resolution. The measurements collected during PaCE 2022 offer valuable insights into aerosol-cloud interactions and cloud evolution in subarctic cloud systems. This dataset is suitable for researchers in cloud microphysics, atmospheric science, and climate modeling, as well as for instrument calibration and validation in future campaigns. The data can also be integrated with complementary concurrent in-situ aerosol, remote sensing, UAV, and balloon-borne observations during PaCE 2022 to provide a more comprehensive understanding of cloud microphysics and atmospheric processes in the subarctic environment. The dataset is publicly available here: https://doi.org/10.5281/zenodo.15045295, Doulgeris et al.,2025.

## 1 Introduction

Clouds play a key role in Earth's climate system, influencing radiative transfer, hydrological processes, and atmospheric dynamics (Boucher et al., 2013; Sherwood et al., 2020). The Arctic region exerts a significant influence on global climate change due to its rapid warming, which is occurring approximately four times faster than the global average, a phenomenon known as the Arctic amplification effect (Post et al., 2019, Rantanen et., al 2022). The impacts of Arctic amplification are expected to extend far beyond the region itself, influencing global weather patterns and climate systems (Shupe et al., 2022). Low-level clouds play a crucial role in the Arctic climate by contributing to the warming of near-surface air, primarily through their interaction with longwave radiation (Shupe and Intrieri, 2004; Zuidema et al., 2005; Maillard et al., 2021). The



unique meteorological conditions of arctic regions, characterized by low temperatures and frequent mixed-phase cloud occurrences, present a distinct challenge in understanding their microphysical and radiative properties (Wendisch et al., 2019). Despite their importance, in-situ measurements of these clouds remain sparse, leading to significant gaps in understanding their behavior, dynamics, and climatic impacts (Baumgardner et al., 2017). Aerosol-cloud interactions (ACI)

play a key role in shaping cloud microphysics, governing processes such as droplet nucleation, ice formation, and precipitation development, which are critical for accurate climate modeling (Morrison et al., 2020). As polar regions are experiencing some of the fastest rates of warming globally, understanding these cloud processes is vital for improving climate projections, particularly in the context of polar amplification (Rantanen et al., 2022).

The Pallas Cloud Experiment (PaCE) 2022 was designed to address these gaps through an integrated,
multidisciplinary approach. Conducted in the Finnish subarctic, PaCE 2022 combined ground-based, airborne, and remote sensing observations to examine aerosol-cloud interactions under subarctic conditions. Among the several instruments employed, cloud spectrometers played a key role in characterizing the microphysical properties of low-level clouds, providing high-resolution measurements of cloud microphysical properties. In situ ground-based measurements using cloud spectrometers are fundamental, as they provide detailed access to individual cloud droplets within a defined sampling
volume, enabling precise characterization of cloud microphysical properties (Wandinger et al., 2018; Doulgeris et al., 2023). This study focuses on the deployment and data collected from four cloud spectrometers during PaCE 2022: the Forward Scattering Spectrometer Probe (FSSP-100), the Cloud, Aerosol, and Precipitation Spectrometer (CAPS), the Cloud Droplet Analyzer (CDA), and the ICEMET holographic sensor. By presenting and analysing these datasets, this work contributes to new insights into the microphysical characteristics of subarctic clouds and their implications for regional and global climate
systems. These measurements offer a resource for improving parameterizations of aerosol-cloud processes in climate models and addressing uncertainties in cloud radiative effects.

This paper specifically details the operation of these cloud spectrometers, and the calibration procedures employed. A comprehensive overview of the Pallas Cloud Experiment 2022 (PaCE 2022) campaign is provided in Brus et al. (2025). The data presented here aims to enhance the understanding of aerosol-cloud interactions and cloud microphysics in the
subarctic, contributing a valuable dataset to support future research in atmospheric and climate sciences.

## 2 Methods

### 2.1 Measurement site

The measurement site is located at the Sammaltunturi station, which is part of the Pallas Atmosphere-Ecosystem Supersite in Finnish Lapland, operated by the Finnish Meteorological Institute (FMI). The station is located approximately 170 km north
of the Arctic Circle (67.9733° N, 24.1157° E, 565 meters above sea level). The surrounding area is characterized by low-lying vegetation, primarily consisting of lichen, moss, and small vascular plants, while the forest below the station is dominated by pine, spruce, and birch trees. This remote location, situated within the Pallas-Yllästunturi National Park,





experiences minimal anthropogenic influence, providing an ideal setting for monitoring natural atmospheric conditions. During autumn, the station is frequently immersed in clouds, offering an optimal environment for studying cloud droplets, ice crystals, and associated aerosol particles. A full description of the measurement site can be found in Lohila et al., (2015), Doulgeris et al., (2022).

**2.2 Instrumentation**

The PaCE 2022 campaign at the Sammaltunturi station utilized an extensive array of meteorological and cloud spectrometry instruments to measure cloud microphysical properties and meteorological parameters. The following section describes the spectrometers and their roles in the campaign. Table 1 provides an overview of the cloud spectrometers employed in the study, detailing the measurement range, measured parameters, derived parameters, and sampling frequency for each instrument while Fig. 1 illustrates their deployment configuration at the Sammaltunturi station.

The Cloud, Aerosol, and Precipitation Spectrometer (CAPS), deployed during the PaCE 2022 campaign, is an instrument designed to measure a wide range of cloud and aerosol microphysical properties. CAPS consists of three instruments, including the Cloud and Aerosol Spectrometer (CAS), the Cloud Imaging Probe (CIP), and a hot-wire Liquid Water Content (LWC) sensor. During PaCE 2022 only CAS was operational. While CAPS has been widely used for airborne measurements (Baumgardner et al., 2001, Spanu et al.,2020), its ground-based adaptation at the Sammaltunturi station faced operational challenges, particularly with the hot-wire LWC sensor, which was prone to ice accretion in supercooled liquid cloud conditions. Also, the CIP was not operational during PaCE 2022 due to technical issues. The CAS utilizes the Mie scattering theory to calculate particle size based on light intensity. The CAPS spectrometer was fixed to one direction that was the main wind direction of the station (220°) to ensure that most of the time would face the wind.

The Forward Scattering Spectrometer Probe (FSSP-100, model SPP-100, DMT, Brenquier, 1989), originally designed for airborne applications, was adapted for ground-based use at the Sammaltunturi station, where it was installed on a rotating platform to ensure it continuously faced the wind. Despite modifications and anti-icing systems, the instrument faced challenges with snow and ice accumulation, which required regular maintenance. Both the CAPS and FSSP-100 were equipped with high-flow aspiration systems to maintain constant flow through the inlets, with air speed monitored and adjusted regularly. Both CAS and FSSP-100 operate on the same principle. They function by detecting forward-scattered light from individual droplets as they pass through a laser beam. The droplet sizes are derived using Mie scattering theory, with backscatter optics also utilized to measure light intensity in the range of 168° to 176°, which helps determine the refractive index of spherical particles. For the calibration of the CAS and FSSP-100, glass beads in the diameter size range 2–40 µm and polystyrene latex sphere (PSL) standards in the diameter size range 0.74–2 µm were used. A detailed description of both instruments and their ground setup can be found in detail in Doulgeris et al., (2020).

The Cloud Droplet Analyzer (CDA, Palas GmbH) is a high-resolution optical aerosol spectrometer specifically designed to measure the size distribution and number concentration of aerosols and cloud droplets. This aerosol spectrometer



can measure the size of dust particles and, under appropriate conditions, cloud droplets, while also determining the water content of the air. The central element of the device is an optical aerosol sensor that uses Mie scattered light analysis of single particles to determine their size. Each particle passes through an optically differentiated measurement volume, which is uniformly illuminated by a polychromatic LED light source. As particles move through this volume, they generate scattered light pulses, which are detected at an angle between 85° and 95°. The number of scattered light pulses is used to determine the particle count, while the intensity of the scattered light pulse serves as a measure of the particle diameter. The CDA has a comprehensive measurement range spanning from 0.2 µm to 100 µm. However, it can only measure particle concentrations from 0 to 200 particles cm$^{-3}$. To ensure efficient particle sampling, a volume flow rate of 5.0 l.min$^{-1}$ should be maintained. The CDA operates with a power consumption of 200 W and is resilient in ambient temperatures ranging from -20°C to +50°C. The device weighs 35 kg and has dimensions of 883 x 640 x 390 mm. During the campaign it covered a size range from 0.4 to 94 µm using 38 bins (in terms of water equivalent diameters). The sampling system includes a vertical inlet tube measuring 15 cm in length, with an inner diameter of 8 mm and an outer diameter of 10 mm. The sampling head is a Sigma inlet. The calibration of the CDAs was carried out before and after the campaign using MonoDust 1500, a dust with a specified and known particle size.

The ICEMET sensor (Kaikkonen et al., 2020), deployed during the PaCE 2022 campaign, was installed on a measurement platform mounted on a ~2-meter-tall pole with a 60 mm diameter at the Sammaltunturi station roof top. uses digital in line holographic imaging to measure droplets and ice crystal properties in cloud samples. The sensor is capable of measuring particles ranging from 5 to 200 µm in diameter, with a maximum particle size of approximately 1 mm. For the PaCE data, the size range was limited to 5–70 µm to better isolate ice crystals from liquid water content (LWC) and median volume diameter (MVD) data.

The particle sizing of the ICEMET sensor at Sammaltunturi station was calibrated on 18[th] February 2021 using NIST tracable monodisperse glass beads (Whitehouse scientific Ltd) with the sizes 9.18 µm and 25.60 µm, and non-tracable 5 µm glass beads. Holographic imaging systems typically overestimate the smallest particle sizes close to the optical resolving power limit of the system (Henneberger et al. 2013). To address this overestimation of smallest particle sizes, a linear correction curve based on the calibration results for all particle sizes under 13.07 µm has been implemented to the particle analysis of ICEMET sensor data. Additionally, the sensors sampling and measurement performance in icing conditions have been recently studied in comparison to the ISO12494:2017 atmospheric icing standard (Molkoselkä et al. 2023).

The maximum hologram recording frame rate of the ICEMET sensor is 6 fps, corresponding to a sampling rate of 1.5 cm³s$^{-1}$. Typically for long term measurements, hologram frame rates of either 0.5 fps or 1 fps are used due to data management reasons. At the Sammaltunturi station, the ICEMET sensor was set to an averaged sampling speed of 0.13 cm³s$^{-1}$, which corresponds to 7.5 cm³min$^{-1}$. To prevent ice formation on the sensor, it is equipped with an anti-icing heating system, operating at 500 W. The sensor runs on a 48 VDC power supply.





130       For the CAPS and FSSP-100, data acquisition software provided by Droplet Measurement Technologies (PADS 2.5.6) was employed. For the CDA the same equations for the derived parameters were adopted. The parameters obtained included number concentration ($N_c$), liquid water content (LWC), median volume diameter (MVD), and effective diameter (ED). The equations are described in detail in e.g. Doulgeris et al., (2020). Uncertainties are estimated at approximately 20% for droplet sizing and 16% for number concentration (Baumgardner, 1983; Dye and Baumgardner, 1984; Baumgardner

135 et al., 2017). Coincidence errors, dead-time losses, and velocity acceptance ratio (VAR) uncertainties were not considered significant during the campaign, as the majority of droplet number concentrations were below 300 cm⁻³. However, uncertainties in LWC measurements remained high, with an estimated margin of error of 40%, consistent with previous studies (Droplet Measurement Technologies Manual, 2009).

      Considering the ICEMET sensor, it was operated using a hologram analysis software ICEMET server that

140 performed several critical functions (Molkoselkä et al., 2021). The software reconstructed holograms, segmented particles within them, autofocused the identified particles, and generated binary images of the focused particles. All single-particle data were stored in a comprehensive database. For the calculation of liquid water content (LWC) and mean volume diameter (MVD), the software applied filtering criteria to the particle data based on preset settings. Specifically, particles were included if they had an effective diameter between 5 and 200 µm and a Heywood roundness value of less than 1.2, where 1

145 represents a perfect sphere. To ensure reliable measurements, the analysis volume was limited to a region 5 mm from the protective windows of the ICEMET instrument, minimizing potential biases and ensuring more isokinetic sampling (Juttula et al., 2022). These filtering steps enabled the compilation of accurate time-series data for cloud microphysical parameters.

Table 1: Summary of instrumentation used for cloud measurements

| Instrument | Measurement Range | Measured Parameters | Derived Parameters | Sampling frequency/ inlet orientation |
|---|---|---|---|---|
| FSSP-100 | 2- 47 µm | Droplet size distribution, | Number concentration ($N_c$), median volume diameter (MVD), liquid water content (LWC) | 1hz/horizontal |
| CAS | 0.61-50 µm | Droplet size distribution, | Number concentration ($N_c$), median volume diameter (MVD), | 1hz/horizontal |





| | | | liquid water content (LWC) | |
|---|---|---|---|---|
| CDA | 0.4- 94 µm | Droplet size distribution, | Number concentration ($N_c$), median volume diameter (MVD), liquid water content (LWC) | 1hz/vertical |
| ICEMET | 5- 200 µm | Droplet size distribution, | Number concentration ($N_c$), median volume diameter (MVD), liquid water content (LWC) | 1hz/horizontal |



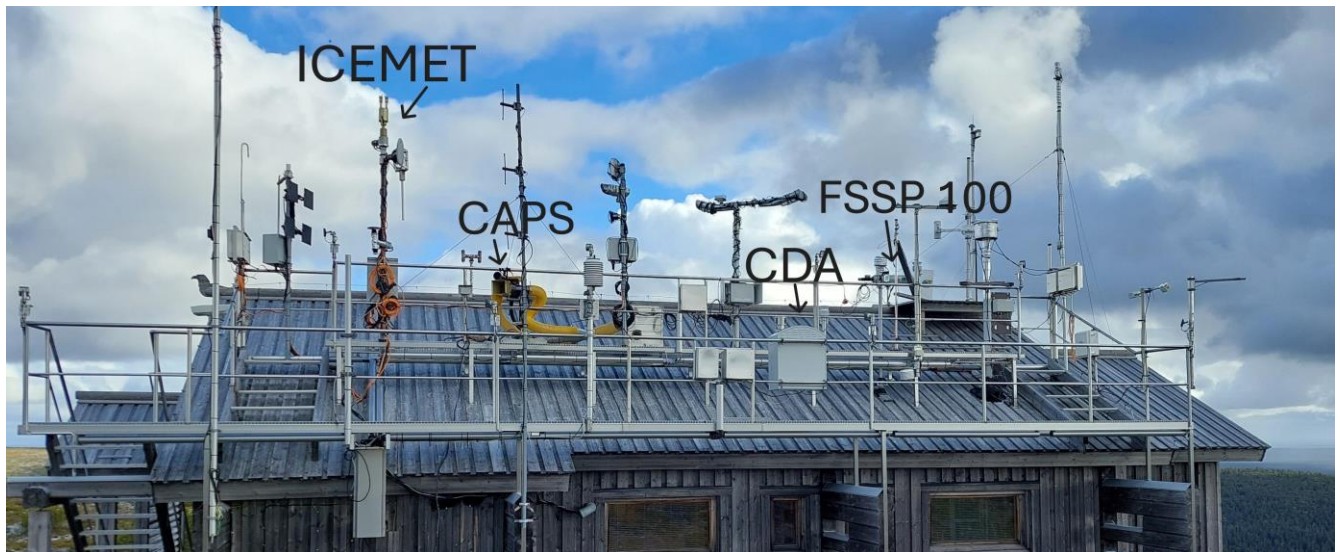

Figure 1. The ICEMET, the CAPS, the CDA and the FSSP-100 as they were installed on the roof at the Sammaltunturi site during PaCE 2022.

To support cloud microphysics observations, the Sammaltunturi station was equipped with several meteorological instruments that continuously monitored environmental variables during the campaign. The meteorological setup included an automatic weather station (Milos 500, Vaisala Inc.) that recorded key meteorological parameters. Horizontal visibility, essential for cloud and precipitation detection, was measured using a visibility sensor (FD12P, Vaisala Inc.). Relative humidity at the station's elevation of 570 m was monitored with a humidity sensor (HUMICAP, Vaisala Inc.), while barometric pressure readings were provided by BAROCAP sensors (Vaisala Inc.). Ambient temperature was measured with high precision using PT100 sensors. To monitor solar energy, the station included radiation instruments, with a pyranometer measuring global radiation and a photoelectric detector recording photosynthetically active radiation (PAR). Wind conditions were documented using a heated cup anemometer to capture wind speed and a heated wind vane to record wind direction, both of which were designed to function effectively under the challenging subarctic conditions. Meteorological measurements were logged as 1-minute averages to ensure high temporal resolution. These data provided critical contextual information on the prevailing atmospheric state, complementing the observations of cloud microphysics. All the weather sensors utilized in this study were previously detailed in Hatakka et al. (2003).

## 3 Quality control and data set evaluation

The current dataset contains only in-cloud measurements, taken when the station was immersed in a cloud. However, we do not explicitly classify the cloud phase (liquid, ice, or mixed phase) associated with these events. Data from each cloud probe





and the weather station were quality-controlled and harmonized into a common format for release and further analysis. The
presence of a cloud at the station was identified using three different factors. First, we checked the droplet size distribution
measured by each of the cloud spectrometers. This was the primary parameter used to determine that the station was inside a
cloud. To confirm this, we cross-checked the droplet counts with two meteorological variables: the relative humidity at the
measurement site, which was expected to be approximately 100%, and the horizontal visibility, which should have been less
than 1 km when the Sammaltunturi station was inside a cloud. In cases where the visibility sensor was not functioning
properly, but the other criteria indicated cloud presence (e.g., droplet size distribution and relative humidity), those
measurements were still included in the dataset. Additionally, in instances where the other criteria confirmed cloud presence,
but the visibility was slightly over 1 km, these data points were also retained in the dataset. To ensure consistency, we
required that these criteria should be met continuously for 30 minutes. If one of these factors was not satisfied, a final
inspection was conducted visually using pictures recorded by an automatic weather camera installed on the station's roof.

The measurements were further inspected to ensure the data set's quality. First, the raw data set was checked to
eliminate any cases where one of the cloud probes was partially or fully blocked. Blocked probes were identifiable from the
raw data. To detect blocked probes, the droplet number concentration ($N_c$) was carefully examined across the entire dataset.
A sudden decrease in $N_c$, followed by a rapid increase, indicated probe inlet freezing. This behavior was attributed to the
probe inlet becoming obstructed by ice or snow, resulting in a higher air speed through the probe. Since the data analysis
assumed constant probe air speed, this abnormality was corrected. The necessary adjustments, as outlined by Doulgeris et al.
(2020), should be under consideration regarding the data from the CAS and FSSP-100 setups. Doulgeris et al. (2020)
demonstrated that the CAS, which was fixed in one direction, showed significant sampling losses when not facing the wind
direction, as it was not sampling iso-kinetically. All in-cloud data for CAS was included in the dataset, but users should
crosscheck CAS data against wind direction to ensure reliability. We recommend using only CAS data collected while
aligned with the wind direction, within the range of 190° to 250°. Missing data points were marked as −9999.9.

The sampling time for all instruments, including CAS, FSSP-100, CDA, and ICEMET, was set to 1 second (1 Hz).
1-minute averages were calculated for each cloud spectrometer. The meteorological parameters were measured by PT100
sensors, Vaisala HUMICAP, BAROCAP sensors, pyranometer, heated cup and wind vane and data were saved as one
minute averages. The FD12P Vaisala weather sensor had a 15-second sampling time. This resulted in the cloud droplet size
distribution and various meteorological variables being available for each minute, as well as derived parameters such as $N_c$
($cm^{-3}$), LWC (g $cm^{-3}$), MVD (µm), and ED (µm). All datasets were converted to both NetCDF and CSV formats. Times are
given in UTC.

The dataset includes separate NetCDF and CSV files for each cloud spectrometer under the file names as described
in Brus et al., (2025). In our case, each file specifically provides data for every month when clouds were present, with the
date in the file name representing the starting date of that month. Each cloud instrument has a separate file (e.g.,
FMI.CAPS.b1.20221201.nc and FMI.CAPS.b1.20221201.csv). Each metadata file contains the sampling area (mm²) and
probe air speed (ms⁻¹) used to derive each parameter. Each file includes the cleaned timeline of the following cloud



properties and meteorological variables: year (YYYY), month (MM), day (DD), hour (HH), minute (MN), number concentration (cm$^{-3}$), liquid water content (g m$^{-3}$), median volume diameter (µm), effective diameter (µm), calculated

dN/dlogDp (cm$^{-3}$) values in each bin, temperature at 570 m (°C), dew point (°C), humidity at 570 m (%), pressure (hPa), wind speed (ms$^{-1}$), horizontal wind direction (degrees), global solar radiation (Wm$^{-2}$), photosynthetically active radiation (µmolm$^{-2}$s$^{-1}$), and horizontal visibility (m).

Each instrument's measurements were processed separately, and data quality was carefully ensured for all four spectrometers: CAS, FSSP-100, CDA, and ICEMET. The CAS contains 30 size bins with a forward-scattering upper bin size

of 0.61, 0.68, 0.75, 0.82, 0.89, 0.96, 1.03, 1.1, 1.17, 1.25, 1.5, 2, 2.5, 3, 3.5, 4, 5, 6.5, 7.2, 7.9, 10.2, 12.5, 15, 20, 25, 30, 35, 40, 45, and 50 µm.  the FSSP-100  was set up to use 30 size bins with a forward-scattering upper bin size of 3.0, 4.5, 6.0, 7.5, 9.0, 10.5, 12.0, 13.5, 15.0, 16.5, 18.0, 19.5, 21.0, 22.5, 24.0, 25.5, 27, 28.5, 30.0, 31.5, 33.0, 34.5, 36.0, 37.5, 39.0, 40.5, 42.0, 43.5, 45.0, and 47.0 µm. The CDA contains 38 size bins with a forward-scattering upper bin size of 0.2, 0.3, 0.4, 0.5, 0.6, 0.7, 0.9, 1.1, 1.3, 1.5, 1.8, 2.1, 2.4, 2.8, 3.3, 3.8, 4.5, 5.2, 6.0, 7.0, 8.1, 9.4, 10.9, 12.6, 14.6, 16.9, 19.5, 22.6, 26.1, 30.1,

34.8, 40.3, 46.5, 53.7, 62.1, 71.8, 82.9, 95.8 µm while the ICEMET contains 1-micron equivalent size bins ranging from 6 µm to 200 µm.

## 4 Overview of data set

The dataset includes in-cloud measurements obtained from four cloud spectrometers alongside several meteorological parameters. The Pallas Cloud Experiment (PaCE 2022) campaign took place from 12 September to 15

December 2022, during which the station experienced cloud conditions—defined as visibility below 1000 m—for 49.5% of the campaign duration. To measure cloud microphysical properties, the following spectrometers were deployed: the Cloud, Aerosol, and Precipitation Spectrometer (CAPS), the Forward Scattering Spectrometer Probe (FSSP-100), ICEMET, and the Cloud Droplet Analyzer (CDA). The CAPS, FSSP-100, and ICEMET were operational throughout the entire campaign period, while the CDA was active from 12 September to 30 November 2022, after which it was relocated to the Sonnblick

observatory for the ACTRIS RI Cloud In Situ European Centre for cloud ambient Intercomparison (CIS ECCINT) Campaign.

During their operational periods, the instruments achieved different levels of data coverage. The CAS achieved a coverage of 87.7%, with data losses primarily caused by clogging of the inlet during cloud conditions. The FSSP-100 had 72.3% coverage, with interruptions mainly due to technical issues. CDA showed the highest coverage at 99.5%, reflecting

nearly continuous operation during its deployment period. In contrast, ICEMET achieved 73.9% coverage, with occasional interruptions caused by minor technical difficulties. Despite these challenges, the dataset remains robust, providing reliable measurements of cloud droplet concentrations, size distributions, and liquid water content under subarctic cloud conditions.

The boxplots presented in Figure 2 highlight the monthly variability of key meteorological parameters during periods of reduced visibility (< 1000 m), with at least one cloud spectrometer operational. Panel (a) shows the temperature at



570 meters above ground level (°C), which decreased steadily from September to December. The average temperature was –1.8°C in September (ranging from –18.2°C to 8.9°C), 0.6°C in October (-5.1°C to 6.4°C), -4.2°C in November (-13.3°C to 2.6°C), and –8.2°C in December (-18.2°C to –4.2°C), with the lowest values observed in December. Panel b) illustrates the wind speed (ms⁻¹) with averaged values approximately 8 ms⁻¹ for each month. Panel c) presents the FD12P visibility (m), which remains consistently below 1000 m throughout the campaign when the station was inside a cloud, with pronounced

reductions in October and December compared to September and November respectively. The median visibility for each month was below 400 m. The average visibility values were 278 (SD 202) m in September, 208 (SD 159) m in October, 328 (SD 200) m in November, and 335 (SD 236) m in December. These monthly trends indicate progression toward colder and windier conditions, increasing the challenges for cloud and atmospheric measurements during the late campaign period.



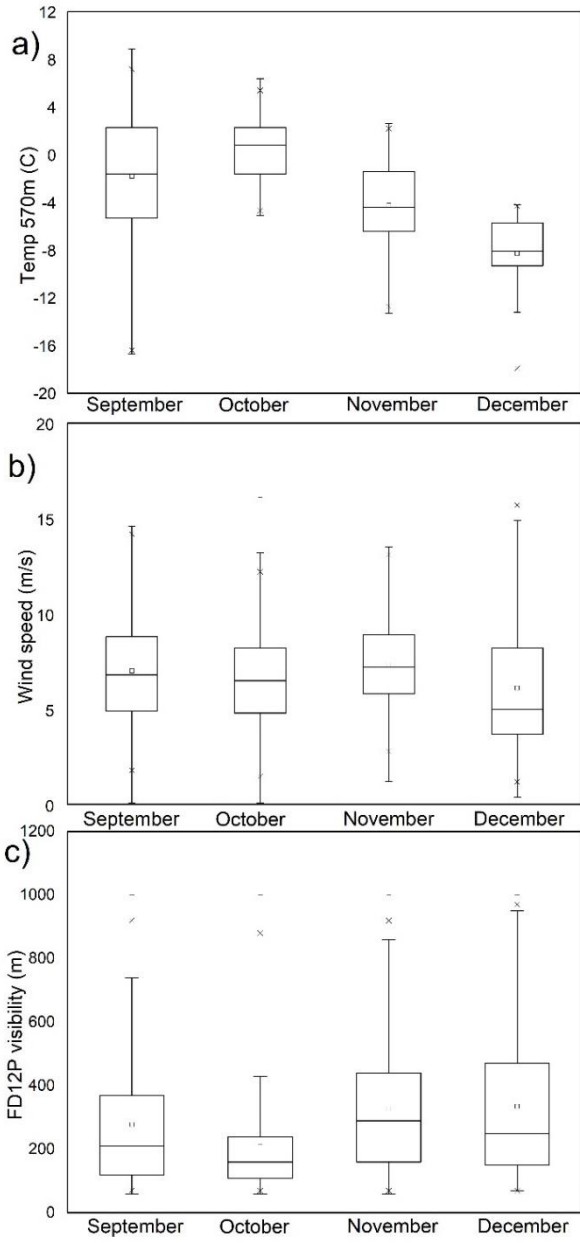

Figure 2: Statistical description of temperature at 570 m above ground level (a), wind speed (b), and FD12P visibility (c) during the campaign period (September–December), considering only periods when visibility was less than 1000 m (i.e., the station was in cloud) and at least one cloud spectrometer was operational. The boxplots depict the median, interquartile range (IQR), and outliers.

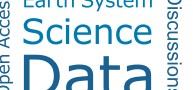



Figure 3 presents the time-series of the cloud droplet number concentration ($N_c$) measured by the four cloud spectrometers. Panel a) shows measurement from the CAS probe, which sampled in a fixed direction (220°). The variations observed in $N_c$ can be partially attributed to the fixed sampling orientation, which may result in lower values when cloud conditions were not aligned with the instrument's sampling direction, for details see Doulgeris et al, (2020). Panel b) displays $N_c$ measured by the FSSP-100, which follows the wind

direction. As a result, the FSSP-100 exhibited relatively consistent measurements with minimal losses. Panel c) shows $N_c$ from the CDA, which employed vertical sampling (perpendicular to wind direction). This method was prone to larger measurement losses, as reflected in the lower $N_c$ values and greater fluctuations compared to the other instruments. Panel d) presents the ICEMET data, which, like the FSSP-100, followed the wind direction and showed consistent measurements with fewer interruptions. The variability across instruments highlights the

influence of sampling orientation and technique on the interpretation of cloud microphysical properties.

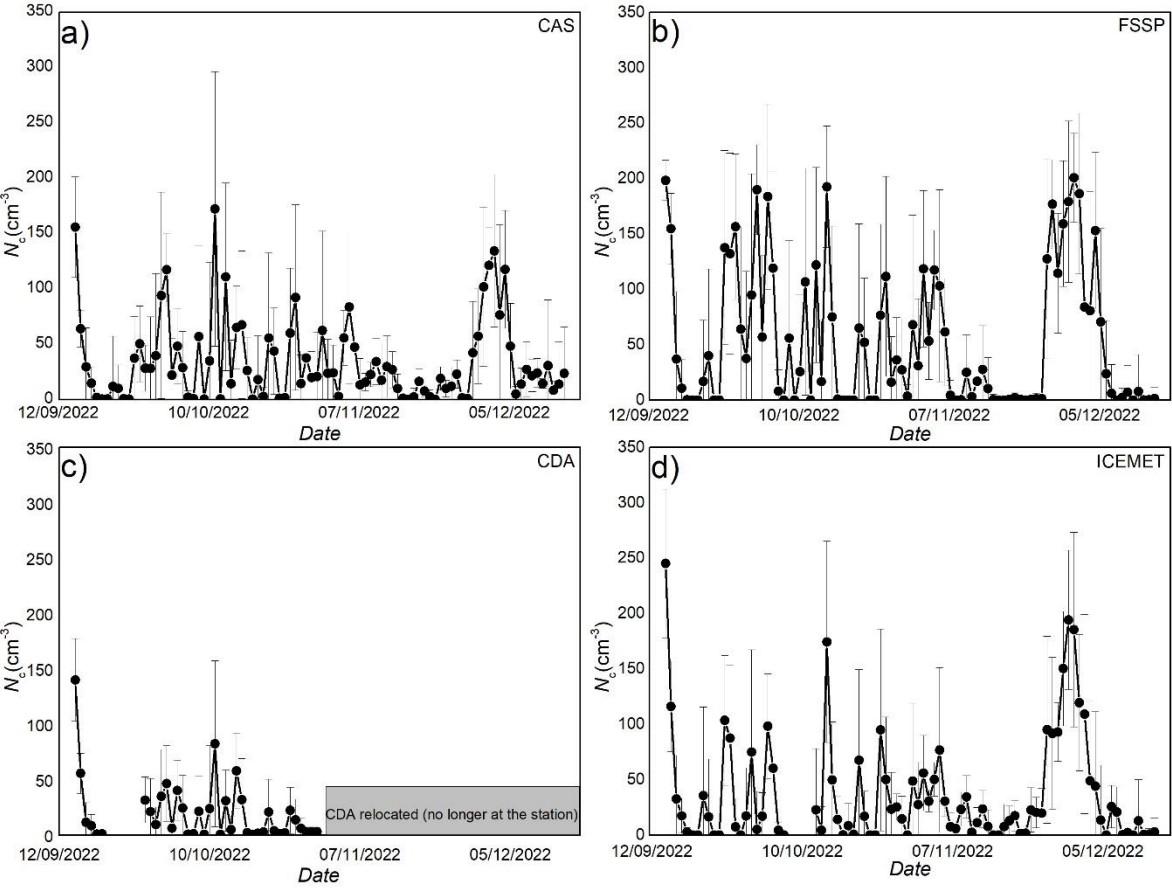

Figure 3: Time-series of daily averaged cloud droplets number concentration ($N_c$) measured by the CAS (a), FSSP (b), CDA (c), and ICEMET (d). Error bars represent the standard deviation (STDV) of daily averages. The CAS, with its fixed orientation, exhibited greater variability, while the FSSP-100 and ICEMET, which align with the wind direction, show more stable $N_c$ values. CDA's vertical sampling resulted in greater losses, as seen in the lower measured $N_c$ values.

Figure 4 displays the monthly variation in median volume diameter (MVD) and liquid water content (LWC) as measured by the ICEMET instrument during cloud conditions (visibility < 1000 m) from September to December. Panel a) shows that the minimum MVD was in December and the larger droplets observed in November. Panel b) shows the LWC, which exhibited slight decreases over time, suggesting a potential reduction in cloud liquid water content as the temperature decreases and atmospheric conditions evolved toward winter. Only ICEMET is presented here because its wide particle size range (5–200 µm) minimizes possible cloud droplet losses and ensures a more comprehensive measurement of cloud microphysical properties, making it well-suited for capturing seasonal variations.

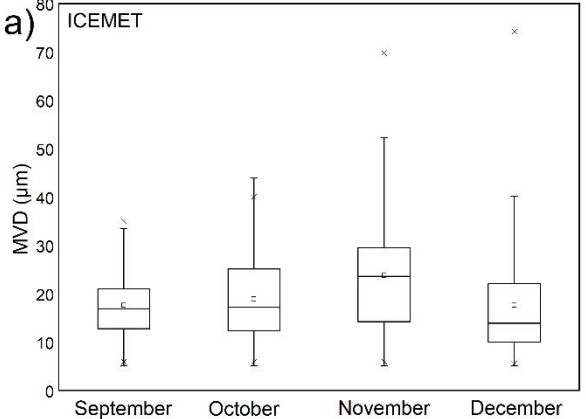
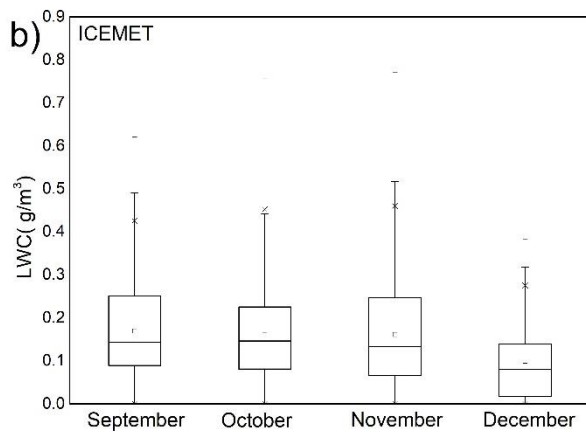

Figure 4: Monthly variations in median volume diameter (MVD, µm) (a) and liquid water content (LWC, gm⁻³) (b) measured by ICEMET during periods of reduced visibility (< 1000 m) from September to December. The boxplots show the median, IQR, and outliers, illustrating a seasonal decline in MVD and LWC.

## 5 Summary

This dataset provides in-cloud measurements collected from four cloud spectrometers: the Cloud Aerosol Spectrometer (CAS), Forward Scattering Spectrometer Probe (FSSP-100), Cloud Droplet Analyzer (CDA), and ICEMET. The data includes droplet size distributions, droplet number concentration ($N_c$), liquid water content (LWC), median volume diameter (MVD), effective diameter (ED) along with key meteorological parameters such as temperature, humidity, wind speed, and



horizontal visibility. All data are provided as 1-minute averages. The dataset spans multiple cloud events, with measurements processed and quality-controlled to ensure reliability.

The dataset is particularly valuable for researchers in cloud microphysics, atmospheric science, and meteorology, offering high-resolution insights into cloud properties and processes under subarctic conditions. Potential end users include atmospheric scientists, modelers, and meteorologists seeking to understand cloud droplet formation mechanisms, aerosol-cloud interactions, and their implications for cloud evolution and climate modeling. Furthermore, the dataset serves as a valuable reference for instrument calibration and validation, particularly in future cloud measurement campaigns.

It is important to note that the sampling characteristics of each instrument play a crucial role in data interpretation. For example, the CAS, with its fixed orientation, may introduce variability in measurements due to directional sampling. The FSSP-100 and ICEMET, which follow the wind direction, provide more consistent data. CDA, while achieving near-continuous data coverage, is more sensitive to vertical sampling losses, which may influence the derived droplet size distributions. Despite these challenges, key sizing parameters (MVD and ED) can still be accurately derived. Special
attention must also be given to LWC estimates, as they are particularly sensitive to instrument performance and environmental conditions (Tiitta et al., 2022). Researchers are encouraged to analyze these parameters within the context of each instrument's sampling limitations to ensure robust conclusions. Also, future studies could benefit from performing a detailed back-trajectory analysis and organizing the dataset based on air mass origins. By integrating these data sources, researchers can achieve a more holistic perspective on cloud processes and their role in regional and global atmospheric
systems. This dataset not only supports fundamental research into cloud physical properties but also has applications in improving climate models, validating remote sensing techniques, and understanding cloud responses to environmental changes. The dataset holds potential as a supplementary tool for improving numerical models, particularly for refining the representation of cloud microphysics in simulations like large eddy models.

**6. Code and data avalaibility**

The datasets are archived with individual DOIs at the Zenodo Open Science data archive as NetCDF and CSV archives: https://doi.org/10.5281/zenodo.15045295, Doulgeris et al.,2025. A dedicated community for the Pallas Cloud Experiment 2022 has been established at (https://zenodo.org/communities/pace2022/, last access: 21 March 2025 where the data files and additional metadata related to the datasets are hosted. Software developed to process and display data from the cloud
ground-based spectrometers is not publicly available and leverages licensed data analysis software (MATLAB). This software contains intellectual property that is not meant for public dissemination.

**7. Author Contributions**



KD and DB conducted the measurements of the CAPS, FSSP, and CDA. KD conducted the data analysis and wrote the manuscript. All authors contributed to the discussion of the data set and reviewed and edited the manuscript. DB and KD prepared and organized the PaCE 2022 campaign. VK, EM, and AM performed the measurements and the data analysis of the ICEMET.

## 8. Competing interests

The authors declare that they have no conflict of interest.

## 9. Special issue statement

This article is part of the special issue:" Data generated during the Pallas Cloud Experiment 2022 campaign". 220


## 10. Acknowledgements

The authors would like to acknowledge Metsa halitus personnel, namely Mirka Hatanpää, for his valuable support during Pallas Cloud experiment 2022.

## 11. Financial support

This work was supported by ACTRIS-Finland funding through the Ministry of Transport and Communications, the Atmosphere and Climate Competence Center Flagship funding by the Research Council of Finland (Grants 337552). This project has also received funding from the European Union, H2020 research and innovation program (ACTRIS-IMP, the European Research Infrastructure for the observation of Aerosol, Clouds, and Trace gases, Grant 871115)

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

Shupe, MD, Rex, M, Blomquist, B, Persson, POG, Schmale, J, Uttal,T, Althausen, D, Angot, H, Archer, S, Bariteau, L,
Beck, I, Bilberry, J, Bucci, S, Buck, C, Boyer, M, Brasseur, Z, Brooks, IM, Calmer, R, Cassano, J, Castro, V, Chu, D, Costa, D, Cox, CJ, Creamean, J, Crewell, S, Dahlke, S, Damm, E, de Boer, G, Deckelmann, H, Dethloff, K, Du¨tsch, M, Ebell, K, Ehrlich, A, Ellis, J, Engelmann, R, Fong, AA, Frey, MM, Gallagher, MR, Ganzeveld, L, Gradinger, R, Graeser, J, Greenamyer, V, Griesche, H, Griffiths, S, Hamilton, J, Heinemann, G, Helmig, D, Herber, A, Heuze´ , C, Hofer, J, Houchens, T, Howard, D, Inoue, J, Jacobi, H-W, Jaiser, R, Jokinen, T, Jourdan, O, Jozef, G, King, W, Kirchgaessner, A,
Klingebiel, M, Krassovski, M, Krumpen, T, Lampert, A, Landing, W, Laurila, T, Lawrence, D, Lonardi, M, Loose, B, Lu¨pkes, C, Maahn, M, Macke, A, Maslowski, W, Marsay, C, Maturilli, M, Mech, M, Morris, S, Moser, M, Nicolaus, M, Ortega, P, Osborn, J, Pa¨tzold, F, Perovich, DK, Peta¨ja¨,T, Pilz, C, Pirazzini, R, Posman, K, Powers, H, Pratt, KA, Preußer, A, Que´le´ ver, L, Radenz, M, Rabe, B, Rinke, A, Sachs, T, Schulz, A, Siebert, H, Silva, T, Solomon, A, Sommerfeld, A, Spreen, G, Stephens, M, Stohl, A, Svensson, G, Uin, J, Viegas, J, Voigt, C, von der Gathen, P, Wehner, B, Welker, JM,
Wendisch, M, Werner, M, Xie, ZQ, Yue, F. Overview of the MOSAiC expedition: Atmosphere, 2022.

Tiitta, P., Leskinen, A., Kaikkonen, V. A., Molkoselkä, E. O., Mäkynen, A. J., Joutsensaari, J., Calderon, S., Romakkaniemi, S., and Komppula, M.: Intercomparison of holographic imaging and single-particle forward light scattering in situ measurements of liquid clouds in changing atmospheric conditions, Atmos. Meas. Tech., 15,
2993–3009, https://doi.org/10.5194/amt-15-2993-2022, 2022.



Wandinger, U., Apituley, A., Blumenstock, T., Bukowiecki, N., Cammas, J.-P., Connolly, P., De Mazière, M., Dils, B., Fiebig, M., Freney, E., Gallagher, M., Godin-Beekmann, S., Goloub, P., Gysel, M., Haeffelin, M., Hase, F., Hermann, M., Herrmann, H., Jokinen, T., Komppula, M., Kubistin, D., Langerock, B., Lihavainen, H., Mihalopoulos, N., Laj, P., Lund Myhre, C., Mahieu, E., Mertes, S., Möhler, O., Mona, L., Nicolae, D., O'Connor, E., Palm, M., Pappalardo, G., Pazmino, A., Petäjä, T., Philippin, S., Plass-Duelmer, C., Pospichal, B., Putaud, J.-P., Reimann, S., Rohrer, F., Russchenberg, H., Sauvage, S., Sellegri, K., Steinbrecher, R., Stratmann, F., Sussmann, R., Van Pinxteren, D., Van Roozendael M., Vigouroux C., Walden C., Wegene R., and Wiedensohler, A.: ACTRIS-PPP Deliverable D5.1: Documentation on technical concepts and requirements for ACTRIS Observational Platforms, available at: https://www.actris.eu/sites/default/files/Documents/ACTRISPPP/Deliverables/ACTRISPPP_WP3_D3.1_ACTRISCostBook.pdf (last access: 8 December 2024), 2018.

Wendisch, M., Macke, A., Ehrlich, A., Lüpkes, C., Mech, M., Chechin, D., Dethloff, K., Velasco, C. B., Bozem, H., Brückner, M., Clemen, H.-C., Crewell, S., Donth, T., Dupuy, R., Ebell, K., Egerer, U., Engelmann, R., Engler, C., Eppers, O., Gehrmann, M., Gong, X., Gottschalk, M., Gourbeyre, C., Griesche, H., Hartmann, J., Hartmann, M., Heinold, B., Herber, A., Herrmann, H., Heygster, G., Hoor, P., Jafariserajehlou, S., Jäkel, E., Järvinen, E., Jourdan, O., Kästner, U., Kecorius, S., Knudsen, E. M., Köllner, F., Kretzschmar, J., Lelli, L., Leroy, D., Maturilli, M., Mei, L., Mertes, S., Mioche, G., Neuber, R., Nicolaus, M., Nomokonova, T., Notholt, J., Palm, M., van Pinxteren, M., Quaas, J., Richter, P., Ruiz-Donoso, E., Schäfer, M., Schmieder, K., Schnaiter, M., Schneider, J., Schwarzenböck, A., Seifert, P., Shupe, M. D., Siebert, H., Spreen, G., Stapf, J., Stratmann, F., Vogl, T., Welti, A., Wex, H., Wiedensohler, A., Zanatta, M., and Zeppenfeld, S.: The Arctic Cloud Puzzle: Using ACLOUD/PASCAL Multiplatform Observations to Unravel the Role of Clouds and Aerosol Particles in Arctic Amplification, B. Am. Meteorol. Soc., 100, 841–871, https://doi.org/10.1175/BAMS-D-18-0072.1, 2019.

Zuidema, P., Baker, B., Han, Y., Intrieri, J., Key, J., Lawson, P., Matrosov, S., Shupe, M., Stone, R., and Uttal, T.: An Arctic springtime mixed-phase cloudy boundary layer observed during SHEBA, J. Atmos. Sci., 62, 160–176, 2005.