# Peer review of "In situ surface cloud measurements dataset from four cloud spectrometers during the Pallas Cloud Experiment (PaCE) 2022."

_Earth System Science Data, 2025_

## Author Comment (AC1)

We would like to thank the reviewer 1 for the thorough and constructive feedback on our dataset and manuscript. We appreciate the time and effort taken to evaluate the data quality and presentation, and we are pleased that the dataset is recognized as a valuable resource for cloud microphysics and meteorological research. Below can be found responses to reviewers' comments as RC - reviewer comment and AA – authors answer.

**Reviewer 1**: The presented document by Doulgeris et al. (2025) introduces a comprehensive and well-structured dataset of in-situ cloud microphysics measurements collected during the Pallas Cloud Experiment (PaCE) 2022.

The authors provide a clear and detailed overview of the instruments used, including the Cloud Aerosol Spectrometer (CAS), the Forward Scattering Spectrometer Probe (FSSP-100), the Cloud Droplet Analyzer (CDA), and the holographic ICEMET sensor. Particularly noteworthy is the transparent description of instrument characteristics and their respective limitations, such as measurement losses due to icing and alignment issues.

The methodology of data collection and processing, along with accompanying meteorological measurements, is comprehensively described and easy to follow. A central aspect of the document is the detailed quality control, clearly identifying potential sources of error and providing suitable solutions and recommendations for data use, particularly concerning the CAS data due to its fixed orientation.

**Issues:**

**RC1.** The metadata of the individual instruments could be further expanded (e.g., serial number, calibration values, calibration dates, first installation date, etc.).

**AA1** We agree with the suggestion to expand the metadata for the individual instruments. In the revised version of the dataset and manuscript, we have added updated metadata files, including serial numbers, calibration dates, calibration values (where available) and installation dates.

**RC2.** To make the dataset more transparent and easier to interpret for future analyses, I suggest introducing a QA flag. This would support the well-documented quality controls and help reduce potential misinterpretations, particularly with regard to CAS and wind direction. One example: 2.November 11:56 – 15:04 Is this gap caused by icing?

**AA2:**

We appreciate the recommendation to include a QA flag to improve transparency and facilitate future data interpretation. For this reason, we introduce a QA flag system across the datasets, which is described in detail in Section 3 of the revised manuscript. The QA flag identifies questionable or missing data due to known issues such as icing, probe misalignment, or power interruptions. Specifically, the data gap on 2 November from 11:56 to 15:04 has been flagged, and we believe this event is very likely related to probe icing. To systematically identify probe freezing events, we closely examined the droplet number concentration (Nc) time series from the CAS probe across the entire dataset. Freezing cases were typically indicated by a sudden drop in Nc, often preceded by a brief spike. This behavior results from the progressive narrowing of the probe inlet due to ice accumulation, which reduces the actual Probe Area Sampled (PAS). Since the PAS is assumed to be constant during data processing, this reduction causes an overestimation of Nc just before the blockage, followed by an abrupt drop when the inlet becomes significantly obstructed. This pattern in Nc was a consistent and reliable indicator of icing-related measurement errors and has been used as a key criterion for QA flag assignment.

**RC3.** The meteorological data from the individual devices differ — for example, the ICE-MET temperature and wind direction are not the same as those in the CDA dataset. Does the CDA dataset include parameters from its internal weather station? This should be clearly stated in the manuscript, as well as in the metadata and the dataset itself.

**AA3** Thank you for pointing out the differences in meteorological parameters (e.g., temperature and wind direction) between ICE-MET and CDA datasets. CDA dataset *does not* contain meteorological data from its own internal weather station. The meteorological parameters included in the CDA dataset originate from the ICE-MET system. This discrepancy in values may be due to an earlier data merging step or incorrect referencing during data preparation. We rechecked and corrected this inconsistency to ensure that the meteorological data source is clearly and accurately indicated in both the manuscript and the metadata.

**RC4** The ICE-MET dataset contains noticeable LWC outliers that could affect the data when grouped temporally. It is caused by values in the upper bins. Eg. 22. October 3:05 UTC Bin 187 Is there an explanation for that —is it already precipitation?

**AA4** We thank the reviewer for pointing this out. We have investigated the outliers in the upper size bins, such as the case on 22 October at 03:05 UTC (Bin 187). Based on image analysis and shape metrics (Heywood roundness), these particles are likely large ice crystals, specifically hexagonal plates that appear nearly round in shape. Because of their round appearance, they pass the current roundness-based liquid water filtering and are included in the LWC calculation, although they are not droplets.

These events typically occur during mixed-phase cloud conditions, where both small liquid droplets and larger ice particles coexist. In this case, smaller droplets were present in the same frames, confirming a mixed-phase cloud.

While we considered applying a stricter roundness threshold (e.g., 1.1 instead of 1.2) to filter out more of these large crystals, doing so would unintentionally remove a large number of valid small droplets due to resolution limitations. Therefore, instead of changing the filter globally, we have flagged such outliers in the QA column and added a clear explanation in the metadata and documentation.

To support this classification, we conducted a broader inspection and identified specific periods where ice particles (based on >100 µm effective diameter) were observed. These periods are now flagged in the updated dataset as *possible ice crystals*. The flagged intervals include:

- 22 Oct, 19:30 – 11:42
- 6 Nov, 06:53 – 23:44
- 7 Nov, 22:20 – 23:52 (incl. one large droplet at 23:24:24)
- 12 Nov, 11:18 – 13:53
- 24 Nov, 20:31 – 28 Nov, 23:22
- 29 Nov, 17:37 – 17:54
- 8 Dec, 18:25
- 18 Dec, 14:51 – 17:18
- 21 Dec, 18:48 – 19:00
- 21 Dec, 22:10 – 22:14

- 22 Dec, 01:27

- 26 Dec, 02:49 – 07:59

- 29 Dec, 18:04 – 31 Dec, 03:24

- 31 Dec, 04:23 – 04:31

Additionally, we note in the documentation that example images of these flagged particles can be provided upon request by contacting the corresponding author.

We emphasize that these outliers do not represent true liquid water but result from misclassified large ice particles. Users interested in bulk liquid cloud properties can use the QA flags to exclude these values accordingly.

**Reviewer 1** Nevertheless, the dataset presented constitutes an extremely valuable resource for researchers in the fields of cloud physics, climate research, and meteorology. The careful documentation and provision of data, including uncertainties and boundary conditions, enhance reliability and facilitate their use in future studies.

We thank for the helpful feedback and believe that the improvements have significantly enhanced the quality and value of the dataset.

---

## Author Comment (AC2)

We would like to thank the reviewer 2 for the thorough and constructive feedback on our dataset and manuscript. We appreciate the time and effort taken to evaluate the data quality and presentation, and we are pleased that the dataset is recognized as a valuable resource for cloud microphysics and meteorological research. Below can be found responses to reviewers' comments as RC - reviewer comment and AA – authors answer.

**Reviewer 2:** The manuscript "In situ cloud surface measurements dataset from four cloud spectrometers during the Pallas Cloud Experiment (PaCE) 2022" provides information about cloud microphysical data collected during several months with four different in situ cloud probes.

Description of the instrumentation, calibration procedures, operations and related difficulties and challenges are clearly presented. Especially part linked to limitations for individual instruments is really good and useful for potential users.

**RC1** Title: I would like to propose change to "In situ surface cloud measurements....." Current title "cloud surface measurements" is confusing as no cloud surface was measured.

**AA1**: We agree with the reviewer that the current title may lead to confusion. We will revise the title to *"In situ surface cloud measurements..."* to better reflect the nature of the dataset.

**RC2:** CAPS fixed orientation. I would like to propose adding one more "flag" parameter to data file which will include if measurements were from good and bad wind sector as defined by authors.

**AA2:** As already discussed with Reviewer 1, we will add a flag in the data files indicating whether each measurement is within or outside the defined "good wind sector," as described in the manuscript. This addition will help users filter for higher-quality data more easily. Flag usage will be described in section 3.

**RC3**: CAS clearly suffered significant losses due to inlet orientation. What was the reason to install the inlet vertically? Then losses are extremely difficult to quantify.

The fixed orientation of the CAS inlet was primarily due to the physical dimensions of the instrument and the constraints imposed by the manufacturer-recommended setup. The installation followed the configuration provided by the manufacturer, which was also used in previous deployments such as the one described in Doulgeris et al., 2020.

**RC4:** Although this is not research paper I wonder what is the local orographic effect at the site. This is valuable and important information for potential users if they would like to extrapolate the measurements to larger scale or compare with other sites.

**AA4:** Indeed, orographic effects are present at the site and their influence varies depending on the synoptic situation. During certain conditions, clouds are advected very close to the surface and interact directly with the terrain (i.e., classic orographic clouds). At other times, clouds approach the station from above, without being directly forced by the terrain.

To distinguish between these scenarios, we rely on continuous remote sensing measurements from the nearby Kenttärova station, which is equipped with a ceilometer and cloud radar. These instruments provide vertical cloud structure and base height information that help us assess whether cloud formation is orographically driven in each case.

A detailed overview of the remote sensing setup and its application in classifying cloud types during the PaCE 2022 campaign is provided in Tukiainen et al. (2025)

**RC5:** I am curios how was done correction for flow changes due to icing (page 9 lines 208-211). It is not only about changes in airflow in wind tunnel. Icing will also change the flow pattern and increase losses dur to impaction and turbulence.

**AA5:** We thank the reviewer for this comment. We acknowledge that the phrasing in the manuscript was misleading. In reality, no correction was applied to account for changes in flow due to icing. Instead, data segments where icing was evident as described in the manuscript either through observed instrument malfunction, visual inspection or unrealistic microphysical signatures were excluded from the dataset entirely or flagged in the updated dataset.

We will revise the corresponding section in the manuscript to reflect this and avoid any misunderstanding. We also agree with the reviewer that icing affects not only the flow rate but also the flow dynamics, further justifying our decision to exclude such periods rather than attempt unreliable corrections.

**RC6** : Page 9 (lines 233-241). Listing of all size bins is not necessary in the manuscript. This info should be metadata in data repository.

We accept this suggestion and will remove the detailed list of size bins from the main manuscript. This information will be included instead as part of the metadata in the data repository to maintain clarity and conciseness in the text.

**RC7**:Figure 2 does not add to the manuscript much of valuable information and can be reduced to one text paragraph

We thank the reviewer for this suggestion. While we understand the concern, we believe that Figure 2 provides a helpful visual summary of the campaign timeline and instrument availability, which may be useful for users to quickly assess data coverage and operational context. We have revised the figure and its caption slightly to enhance clarity and reduce redundancy with the text, but we would prefer to retain it in the manuscript for the benefit of data users.

**RC8:** Page 14, Line 319: MVD and ED can be accurately derived only for periods when instruments measure properly and did not suffer losses and which cannot be accurately corrected. I am curios what instrument did provide most of the good quality data for the MVD and ED? FSSP-100?

We agree that accurate derivation of MVD (mean volume diameter) and ED (effective diameter) depends on reliable measurements. However, even during periods when some cloud droplets may be lost due to suboptimal probe performance or positioning, MVD and ED can still be meaningfully derived, particularly when the losses affect the full size spectrum rather than being size-selective. In such cases, the shape of the droplet size distribution remains relatively unaffected, and the derived parameters remain robust. This behavior has been discussed in Doulgeris et al. (2020) and similar patterns were observed during the ECCINT Sonnblick intercomparison campaign (manuscript in preparation, to be submitted in 2025).

In our campaign, the most reliable MVD and ED values were obtained when probes were oriented optimally with respect to the wind, minimizing sampling losses. The FSSP-100 and ICEMET probes generally provided the highest-quality data under such conditions.

**Reviewer 2**: Overall this is well written dataset description with very good instrumental section and after some revisions suitable for publication in ESSD.

references:

Doulgeris, K.-M., Komppula, M., Romakkaniemi, S., Hyvärinen, A.-P., Kerminen, V.-M., and Brus, D.: In situ cloud ground-based measurements in the Finnish sub-Arctic: intercomparison of three cloud spectrometer setups, Atmos. Meas. Tech., 13, 5129–5147, https://doi.org/10.5194/amt-13-5129-2020, 2020.

Tukiainen, S., Siipola, T., Leskinen, N., and O'Connor, E.: *Remote sensing measurements during PaCE 2022 campaign*, Earth Syst. Sci. Data Discuss. [preprint], https://doi.org/10.5194/essd-2024-605, in review, 2025.

---

## Editor Decision (ED1)

**Editor's comment**

**September 4, 2025**

The authors present a valuable data set in a concise and clear way. Previous revisions have made improvements, but I do have some minor suggestions and comments to consider before publication.

**1 General comments**

- **Line 15:** The number concentration should be typeset italic, i.e.  $N_c$  should be  $N_c$  (see Copernicus style guide under Mathematical symbols and formulae). This should also be fixed in other occurences.
- **Line 20:** In-situ should not be hyphenated (see Copernicus style guide under Hyphens).
- Line 22: There is a space missing missing in the reference to Doulgeris et al., 2025 and the sentence should also end with a full stop.
- **Line 33:** In-situ should not be hyphenated (see Copernicus style guide under Hyphens).
- Lines 27 & 38: You refer to it in line 27 as "Arctic amplification", while you refer to it as "polar amplification" in line 38. Chose one of those.
- Line 78: You refer to it at line 15 as "liquid water content", while you capitalize it in line 78. In lines 115-116, 132-133, 142-143, Table 1, 235, 262, 297, 300, 305, 311, you also type out LWC and MVD, even thought it has been defined as an abbreviation before. Sometimes it can be beneficial to get the full name again, but I think it is often not necessary, especially for well-known abbreviations such as the liquid water content.
- **Line 104:** There is a full stop in the unit, which should be  $lmin^{-1}$ .
- **Line 106:** There should be a space between number and unit (see Copernicus style guide under Figure content guidelines).
- **Line 115:** You refer to it in line 27 as "Arctic amplification", while you refer to it as "polar amplification" in line 38. Chose one of those.
- **Line 134:** There should be a space between number and unit (see Copernicus style guide under Figure content guidelines). The same is true for other occurences (e.g. line 198, lines 266-267).
- **Line 336:** There is a space missing after "Doulgeris et al.,".
- Line 338: There is a closing parentheses missing after "21 March 2025".

---

## Author Response (AR2)

**Dear Editor,**

We would like to thank you for your careful reading of our manuscript and for the constructive comments. We have addressed all the points you raised:

- The notation of number concentration (*Nc*) has been corrected according to the Copernicus style guide.
- "In situ" is now written without hyphen throughout the text.
- The reference to *Doulgeris et al., 2025* has been corrected (spacing and punctuation).
- Terminology has been unified to consistently use "Arctic amplification."
- The use of abbreviations (LWC, MVD) has been harmonized: the full term is defined once and abbreviations are used consistently, with only a few reintroductions where beneficial for clarity.
- Units have been corrected (e.g., spacing, removal of extra full stop in *l min* -1).
- Other minor typographical issues (spacing, missing parenthesis) have been fixed.

We believe these changes improve the clarity and consistency of the manuscript and thank you again for your valuable feedback.

Sincerely,

Konstantinos Doulgeris on behalf of all co-authors